# Correlation Analysis between Urban Elements and COVID-19 Transmission Using Social Media Data

**DOI:** 10.3390/ijerph19095208

**Published:** 2022-04-25

**Authors:** Ru Wang, Lingbo Liu, Hao Wu, Zhenghong Peng

**Affiliations:** 1Department of Urban Planning, School of Urban Design, Wuhan University, Wuhan 430072, China; wang_ru@whu.edu.cn (R.W.); lingbo.liu@whu.edu.cn (L.L.); 2Center for Geographic Analysis, Harvard University, Cambridge, MA 02138, USA; 3Department of Graphics and Digital Technology, School of Urban Design, Wuhan University, Wuhan 430072, China; wh79@whu.edu.cn

**Keywords:** urban elements, COVID-19 transmission, social media data, urban planning

## Abstract

The outbreak of the COVID-19 has become a worldwide public health challenge for contemporary cities during the background of globalization and planetary urbanization. However, spatial factors affecting the transmission of the disease in urban spaces remain unclear. Based on geotagged COVID-19 cases from social media data in the early stage of the pandemic, this study explored the correlation between different infectious outcomes of COVID-19 transmission and various factors of the urban environment in the main urban area of Wuhan, utilizing the multiple regression model. The result shows that most spatial factors were strongly correlated to case aggregation areas of COVID-19 in terms of population density, human mobility and environmental quality, which provides urban planners and administrators valuable insights for building healthy and safe cities in an uncertain future.

## 1. Introduction

The 2019 novel coronavirus [1,2], which was first reported in Wuhan, the capital of Hubei province, China, evolved into a worldwide public health event. As the main daily living space for residents, the city had a close internal relationship with people’s health [3,4,5] and there was a close correlation between personal activity related to the city’s attributes and the risk of infection [6]. Although pathogens were transmitted through mobile carriers, the urban environment shaped their flow [3]. The mechanism of the urban environment affected the spatial distribution of COVID-19, and characteristics of the area where cases gathered within the city remained unclear, which required detailed intra-urban analysis to reveal the spatial disparities and determinants of the COVID-19 pandemic.

Personal space and activities related to urban attributes were closely related to infection risk [6] at the city level, and existing research on COVID-19 and its impact factors in urban areas can be classified into three major themes, namely, population density, human mobility and environmental quality. First, studies have demonstrated the effect of spatial aggregation of the population on the susceptibility of people to COVID-19 [7,8]. Second, from the perspective of human mobility, since the distribution of public service facilities affected the urban residents’ mobility, the association of public facilities with the spatial variation of COVID-19 distribution should be further observed [9]. Lai et al. mentioned the impact of human mobility caused by urban traffic [7]. You et al. found that the distribution of commercial facilities and medical service facilities was related to the incidence of COVID-19 [6]. ImKampe et al. proved that COVID-19 outbreaks in schools did occur [10]. Patel et al. referred to the specific performance of food supply facilities during the pandemic [11]. Besides, administrative agencies, as a public service facility for people’s daily affairs, were usually ignored in the existing research [12]. Third, despite the different social, cultural, economic, political and environmental backgrounds around the world, residential environmental quality played an important role in curbing the pandemic and fighting diseases [13]. The relevance of poor communities in COVID-19 infection was also emphasized in existing research [14,15]. You et al. suggested that older communities in Wuhan should be renovated and the corridors of public space should be expanded to alleviate overcrowding [6]. In addition, air quality had also been proved to have a significant correlation with the COVID-19 pandemic [16,17,18] and open spaces such as green spaces and waters played a significant role in people maintaining distance and good health [19,20]. Furthermore, in addition to the elements mentioned in the above study, with the comprehensive consideration of the unique attributes of Wuhan (such as having a population with more than a million college students, etc.), seventeen explanatory variables possibly related to the distribution of the COVID-19 in the main urban area were selected.

Our intellectual contributions were threefold. Firstly, most existing studies had explored the impact of risk factors of COVID-19 at the country level [21], but lacked investigations of the spatial variations of the COVID-19 pandemic at the intra-urban scale, especially for Wuhan, the city where the first outbreak of the pandemic was reported. Our empirical explorations of Wuhan’s experiences of COVID-19 sought to inform detailed urban policy and governance for better preparedness and responses to public health crises.

Secondly, the main challenge hampering the research into spatial correlation between pandemics and urban spatial factors was the lack of the pandemic information on a fine spatial scale, such as on the community or kilometer grid scale. Since the pandemic information released by the Wuhan government was only at the administrative region level [6,22,23], there was no access to more detailed geospatial case data. Social media data were widely used in a myriad of studies related to COVID-19 for geotagging possible patients, analyzing the spatiotemporal dynamics of pandemics [8], studying psychological consequences of the disease [24], investigating the public’s response [25], characterizing the propagation of situational information [26], predicting re-outbreaks of the COVID-19 [27], and examining information flow during the pandemic [28]. Compared with the traditional case data, such social media data had the unique benefits of better accessibility and rich and accurate geographic information, and we innovatively used Weibo data to revisit the COVID-19 pandemic during the early stage in detail, which were found to have good sample coverage and were able to represent the spatial results of the natural transmission of COVID-19 in the main urban area of Wuhan to a certain extent [8].

Thirdly, we draw from the literature to develop a framework of urban elements for better detecting the mechanism of spatial factors affecting the spatial transmission of the disease in urban spaces. A myriad of studies focused on the influence of social and economic factors on COVID-19 [6,29], with no focus on the characteristics of the city itself. Most studies focused on the association of the built environment’s attributes with the spread of COVID-19 [30,31] while minimizing the impact of the natural environment in the urban space. Since the urban form generally encompasses a number of physical features and nonphysical characteristics [32], we hence conducted a novel framework including seventeen urban factors in terms of population density, human mobility and environmental quality to investigate the correlation between infectious outcomes and related urban factors.

Given all the above, the reason why we selected Wuhan as the study region can be attributed to the following three points. First, Wuhan was the first seriously impacted city by COVID-19 [8]. Second, the Weibo cases only occurred in Hubei province, and most of them were concentrated in Wuhan. Third, it was confirmed that the Weibo help data could well demonstrate the three phases of COVID-19’s early spread in Wuhan—scattered infection, community spread, and full-scale outbreak [8]—as well as reflecting the results of natural transmission with less intervention during the early stage of the pandemic.

In this study, we collected about 1200 Weibo messages, which included 729 valid records located in the main urban area of Wuhan, so as to represent the spatial variations of the COVID-19 pandemic in the city. Seventeen explanatory variables, from perspectives of population density, human mobility and environmental quality, namely, population, the elderly population, the third ring road, rivers (the Han river and the Yangtze river), markets, third-class hospitals, middle schools, universities, business points of interest (POIs), administration points of interest (POIs), bus stops, metro stations, house prices, age of buildings, air quality index, green spaces, and waters (rivers and lakes) were chosen to identify their influences on the spatial variation of COVID-19 in the main area of Wuhan. Considering the spatial correlation within Weibo COVID-19 cases and influencing factors, respectively, we applied the Euclidean Distance (ED), Kernel Density Estimation (KDE) and Inverse Distance Weighted (IDW) methods to process these elements, and adopted the multiple linear regression model to test the extent to which the severity of infection was impacted by urban spatial indicators. The possible impacts were further discussed so as to provide urban planners and administrators with valuable insights for building healthy and safe cities in a future of uncertainty.

## 2. Study Area and Data

### 2.1. Study Area

Wuhan is a city in Central China with a total area of 8569.15 square kilometers, a permanent resident population of 12.3265 million, and a total economic output of 1561.61 billion yuan, which was ranked among the top ten cities in China by the end of 2020. Wuhan is an important water, land and air transportation hub in China since it is located at the intersection of the golden waterway of the Yangtze River and the artery of the Jingguang railway and hence it has always been known as the “thoroughfare of nine provinces”.

Wuhan’s main urban area (MUA) is located in the center of the city, and it is the main gathering area for urban activities (Figure 1). According to the confirmed data released by the Wuhan Municipal Health Commission, as of 16 March 2022, Wuhan had reported a total of 50,537 confirmed cases, 76.034% of which were located in the main urban area of Wuhan (Table 1). Therefore, MUA was the region where cases were concentrated in Wuhan and the main administrative areas covered by MUA include Jiang’an, Jianghan, Qiaokou, Hanyang, Qingshan, Wuchang, Hongshan and Wuhan Economic Technological Development districts.

The traffic analysis zone (TAZ) based on the street network was the spatial statistical unit for this study. The average size of TAZ was 0.204 km^2^, with the smallest block being 0.007 km^2^ and the largest block being 5.851 km^2^. There were 1681 blocks in total.

### 2.2. Weibo COVID-19 Cases

In the early stage of the COVD-19 pandemic, due to its rapid outbreak, Wuhan issued a message on the closure of the city on 23 January 2020, which further aggravated the public’s panicked sentiment. A large number of people flocked to the hospital, leading to a patient overflow. Sina Weibo, one of the most influential social media platforms in China, opened a help-seeking channel for patients infected with COVID-19 suffering from pneumonia. Patients who could not receive timely treatment could be treated by publishing relevant basic information on the topic.

The Weibo help information was mainly distributed from 3 February to 12 February, and most of the information was posted from February 4th to 8th and stopped updating until the local government took a series of effective measures to supplement medical resources. The research collected about 1200 Weibo messages over this period of time, for which 729 valid records were located in the main urban area of Wuhan and basic information such as the patient’s age, gender, address, date of illness, and the number of family infections were obtained by eliminating invalid and duplicate information. The onset time of patients included in the effective records ranged from 20 December 2019 to 10 February 2020, and most of them were concentrated from 1 January 2020 to 6 February 2020 (Table 2).

The participation rates in the social media helpline among the COVID-infected at the administrative unit level (Table 3) were further obtained by calculating the ratio of infections in the Weibo information to the cumulative confirmed cases. Due to the sudden outbreak of the disease in the early stage, the officially confirmed data at the administrative region level could only be traced back to 5 March 2020, which may cause deviation to a certain extent. The result showed that the participation rates ranged from 1.46% to 2.49%, which could supplement the reliability of data to some extent.

### 2.3. Explanatory Variables

We selected seventeen explanatory variables from the perspectives of population density, human mobility and environmental quality (Table 4) to test the extent to which the severity of infection was influenced by urban spatial indicators.

In Wuhan, the third ring road and the rivers played a crucial role in separating the population distribution in the urban region [33], and hence the third ring road, rivers, population, and the susceptible population together were used as the measure of population density status. The information on the third ring road and rivers (the Han river and the Yangtze river) was provided by the local planning institute of Wuhan. Mobile phone data with age tags were used in this study to obtain the spatial distribution mapping of the population and the elderly population by the number of users that the base station served, which was summarized by matching base stations with the highest call frequency of users to the user ID.

Since the distribution of public service facilities affected the urban residents’ mobility [9], third-class hospitals, middle schools, universities, business facilities, administration facilities, bus stops, and metro stations were included to measure human mobility. In China, the market was the main place where people frequently entered and exited in their daily lives, and the Huanan Market especially was the first place in Wuhan where the pandemic outbreak was identified; hence, it was included as an element that affected human mobility [34,35,36]. Furthermore, the university was also included in the mobility theme since there were 84 universities located in Wuhan by the end of 2019. The POI data of facilities in 2019 were extracted from the Gaode Map API, which contained public service facilities (middle schools, universities, hospitals, administrations, markets, restaurants) and the transportation facilities (street network, bus stations and subway stations) in the main urban area of Wuhan.

In terms of the environmental quality, five variables were identified: house prices, age of buildings, air quality index, green spaces and waters (rivers and lakes). The sales prices and chronological information of residential properties were taken from the Anjuke website, which may partly reflect the quality of the residential environment. The Air Quality Index (AQI) data of 15 stations in 2019 in the main area of Wuhan were obtained from the local government and were used as the regional air quality indicator. Basic land use information such as green spaces and waters were provided by the local planning institute of Wuhan.

The average of the kernel density analysis of Weibo help cases in each TAZ unit was the dependent variable and seventeen processed variables (Table 4) in terms of the population density, human mobility and environmental quality were the explanatory variables, so as to explore the possible correlation between urban variables and COVID-19 transmission.

## 3. Methods

### 3.1. Kernel Density Analysis

The kernel density method was used to calculate the unit density of the measured values of points and line elements within a specified neighborhood, and could intuitively reflect the distribution of discrete measured values in the continuous area [37,38]. It considered the interaction between spatial elements as well as weight values for specific parameters. Kernel density analysis could be used for facility accessibility [39], the incidence of disease [40], regional analysis [41], traffic analysis [42], etc., and hence kernel density interpolation was used to characterize the distribution of COVID-19 cases from Weibo, the population and the elderly population, the accessibility of transportation facilities, educational facilities, commercial facilities and administrative facilities, and water delivery in the main area of Wuhan, since it was considered that these variables served the local urban area. The equation was as follows:(1)$${\hat{f}}_{h}\left(x\right)=\frac{1}{nh}\overset{n}{\underset{i=1}{\sum }}K\left(\frac{x-x_{i}}{h}\right)$$
where *K* was the kernel (a non-negative function), *h* > 0 was a smoothing parameter called the bandwidth, and $x_{i}$ was the sample point.

Due to the different nature of the facilities, the accessibility measurement should be adopted according to the daily use mode. The commonly used spatial Euclidean Distance (ED) was applied in the study to capture the closest distance from residents to the third ring road, rivers (the Han river and the Yangtze river), third-class hospitals [43], markets, and the surrounding green space [44], since it was considered that hospitals, green spaces, and markets served the entire urban area, and the distance from rivers and the third ring road was used to represent the location characteristics of residents.

### 3.2. Inverse Distance Weighted (IDW)

The Inverse Distance Weighted (IDW) method was used for interpolation to estimate unmeasured cell values. It assumed that things that were closer to each other were more similar than those which were farther apart. Greater weights were assigned to the elements which were closest to the measured cells; consequently, the allocated weights changed as an inverse function of the p_th_ power of distance, where power function (p) was a positive real number [45], and the value of *p* was 2. The Inverse Distance Weighted method was usually used to predict air quality parameters [46,47], spatial rainfall distribution [48], surface water quality [49], etc., and hence it was applied to describe the air quality indicator (AQI) in the main urban area in Wuhan.

### 3.3. Multiple Linear Regression

Multiple Linear Regression was used to determine the relationship between a dependent variable (y) and more than one independent variable (x_1_, x_2_, …, x_k_) in the scientific research [50,51,52]. The general form of the multiple linear regression model was given by
(2)$${y=\beta }_{0}{+\beta }_{1}x_{1\;}{+\beta x}_{2}{+\ldots +\beta }_{k}x_{k}+\varepsilon (k\;\leq \;17)$$
where y was the density value of the Weibo population, x_1_, x_2_, …, x_k_ were values of selected explanatory variables, $\beta_{0}$ was the intercept, $\beta_{1}$, $\beta_{2}$, …, $\beta_{k}$ were regression coefficients of factors, and ε was the random error.

## 4. Results

### 4.1. Spatial Variations of Weibo COVID-19 Cases

According to the number of household infectors reported in each record, the result of the kernel density estimation of COVID-19 cases of Weibo data is presented in Figure 2. The spatial distribution of cases showed relatively concentrated regional patterns in riverside (the Yangtze river and the Han river) areas of the Jiang’an, Jianghan, Qiaokou, Wuchang, and Hanyang districts, as well as the eastern area of East Lake High-Tech Development district.

### 4.2. Correlation Analysis

Weibo help data and seventeen influence factors were coded and the numerical statistical results are shown in Table 5. The scatter diagram of variables (Figure 3) showed the correlation between the density of COVID-19 cases from Weibo and the seventeen independent variables, and strong linear trends were determined between distance to the third ring road and hospitals, kernel density of the population, the elderly population, middle schools, bus stops and metro stations, and the age of buildings and the Weibo help cases with an absolute value of the Pearson’s *r* greater than 0.5.

### 4.3. Multiple Regression Model

As there may be collinearities between variables, this study conducted the calculation of linear regression (Table 6), in which the kernel density of the Weibo help cases was the dependent variable and the rest of the seventeen factors were used as independent variables, and the strength of correlation between the dependent variable and each influence elements changed. Fourteen explanatory variables, the Euclidean Distance (ED) to the third ring road, markets, hospitals, rivers, and green spaces, the Kernel Density Estimation (KDE) of the elderly, middle schools, universities, bus stops, metro stations and waters, house prices, age of buildings, and the Inverse Distance Weighted (IDW) value of air quality index (AQI) were obtained through the collinearity test and 5% significance level test, and the factors of the Kernel Density Estimation (KDE) of the population, business facilities and administrations were excluded from the model due to the existence of collinearity and failure to pass the significance test, respectively. Consequently, a linear regression model of fourteen selected factors was further constructed to observe the mechanism of the spatial distribution of COVID-19 and its determinant urban elements (Table 7). In general, the adjusted R-square of the model was 0.790, which indicated that the selected fourteen independent variables’ explanatory abilities for the dependent variable were credible.

The spatial distribution of selected fourteen explanatory variables in TAZ units were shown in Figure 4. Units with the highest elderly population density were concentrated in the core area of the Jianghan, Jiang’an, Wuchang, as well as Qingshan districts. Markets, third-class hospitals, middle schools and transportation (bus stops and metro stations) facilities presented relatively balanced distribution patterns on the whole. Universities in Wuhan were concentrated on the right bank of the Yangtze River. In terms of environmental quality elements, the distribution of green spaces and water spaces (rivers and lakes) were relatively discrete. High air quality indices were found in the Qingshan and eastern Hongshan districts, and it was shown that the air quality in the center area was the best. The core areas of Wuchang and Hongshan had relatively higher house prices. There were clusters of older buildings in the central areas of Wuchang, Hongshan, Jiang’an as well as Jianghan districts.

The visualization of the standardized coefficients of the fourteen factors regression model was shown in Figure 5 to clearly observe the influence mechanism of factors on the dependent variable. Influence factors of the Kernel Density Estimation (KDE) of the elderly population, middle schools, bus stations and metro stations, the Euclidean Distance (ED) to the third ring road and green spaces, and the Inverse Distance Weighted (IDW) value of air quality index (AQI) had positive correlations with the aggregation of COVID-19 infections. Among them, the Kernel Density Estimation (KDE) of bus stops and the elderly population, as well as the Euclidean Distance (ED) to the third ring road determined particularly significant positive effects, with coefficients of 0.321, 0.312 and 0.267, respectively. The Euclidean Distance (ED) to markets, hospitals and rivers, house prices, Kernel Density Estimation (KDE) of universities and waters, as well as the age of buildings had negative correlations with the density of cases within the city, among which, house prices and the Euclidean Distance (ED) to rivers showed stronger negative effects, with coefficients of −0.178 and −0.174, accordingly.

## 5. Discussion

The study aimed to explore the correlation between urban factors and the transmission of COVID-19 based on Weibo COVID-19 data and seventeen spatial indicators generated by multi-source data. Generally, the characteristics of case aggregation areas in the main urban area of Wuhan can be summed up in three features: high population density, high degree of human mobility and relatively poor environmental quality.

### 5.1. Impacts of Urban Factors on COVID-19 Transmission

In the early stage of the pandemic, residential areas located in the city center, which were measured by the distance to the third ring road, and developed regions along the river tended to be more densely populated, and hence the number of infected people was high, which was consistent with the findings of the literature [53] that larger metropolitan areas had a higher infection and mortality rates. The density of the elderly became the positively relevant factor for the spatial distribution of the patients, indicating that the elderly population was susceptible and was at great risk of serious disease and death in the pandemic [54,55]. In Milan, it was proved that elderly populations suffered from very poor accessibility to primary health services during the pandemic [56].

Second, the kernel density of bus stations, metro stations and middle schools became positively related factors to the spatial distribution of the patients, while the Euclidean Distance (ED) to hospitals and markets and Kernel Density Estimation (KDE) of universities were negatively related factors, which could be attributed to the layout of public facilities such as busses, metro stations, middle schools, hospitals and markets which increased human mobility, and the layout of relatively close universities decreased the degree of human mobility, and hence the probability of people-to-person contact was raised or declined [6]. Areas with better hospital access had higher risks of infection. A possible explanation for this is that due to the unbalanced spatial distribution of medical facilities, patients tended to go to large hospitals for treatment, causing a wider range of cross-infections, which made hospitals become high-risk places for the initial spread of COVID-19 [57].

Third, on the one hand, with the high-density development model of cities in recent years, high-rise residential buildings have become the normal form of residence. Consequently, public elevators and other infrastructures may be risk factors in some relatively new communities during the pandemic, while in old communities, the stairs were well ventilated, and hence led to fewer infections. Simultaneously, house prices were negatively correlated with infection density [15]. Communities with high house prices may have better management services and hence could be well protected during the pandemic, which indicated that the social polarization between the rich and the poor was one of the significant inequality factors during the pandemic. On the other hand, the air quality and the accessibility of open spaces such as green spaces and waters also had slight impacts on the variability of the spatial distribution of the disease, and the study considered that these impacts were formed through the decompositions of the virus and long-term potential effects on residents’ immunity [58]. Furthermore, poor air quality may increase the probability of residents suffering from respiratory diseases in winter, thereby raising the possibility of infection after going to hospitals.

### 5.2. Policy Recommendations for Urban Development

Cities are the habitats of residents. The COVID-19 pandemic showed that many cities in the world were insufficiently prepared to deal with the challenge it presented. Weak links in urban materials and infrastructure were revealed. The urban environment played an important role in suppressing the spread of the virus before the pandemic, and cities should further complete emergency planning and response measures to offset future risks after the pandemic [3], as the weaknesses should be resolved before the next pandemic [59]. The study proposed the following targeted urban planning and management recommendations based on the three characteristics of the areas where cases were highest.

First, areas with high-density populations and susceptible populations were more likely to be infected. For high-density urban areas, especially in Asia, city managers should give priority to identifying areas where the population and susceptible populations gather and implement strict controls so as to prevent the further spread of disease during a pandemic [21]. In Wuhan, the results revealed that areas far from the third ring road (with a positive coefficient of 0.267) or close to the Yangtze river (with a negative coefficient of −0.174) had a large number of people; although high-density cities had an efficient agglomeration effect, the risk of high-density population agglomeration should be reduced in future urban construction, such as multi-center construction and rational evacuation planning in urban areas [7]. In addition, the empirical results showed that the elderly have a high degree of positive relationships with the spatial variation of the disease (with a positive coefficient of 0.312) since they were more likely to develop severe illnesses due to their reduced immunity in the pandemic as the susceptible population. Policymakers could plan to provide specialized services by improving the level of services designed for the elderly, so as to promote healthy aging in urban areas [56].

Second, the high accessibility of public service facilities provided convenience for residents, but it also increased the mobility of the population, leading to a high risk of infection. The unfairness of the distribution of public service facilities and resources aggravated the excessive concentration and mobility of local nodes. The empirical results showed that infections were higher in areas close to hospitals, with a coefficient of −0.870, indicating that urban planning and management departments need to strengthen the rational layout of public service facilities and develop a fair allocation of public service resources, such as promoting the further improvement of the three-level diagnosis and treatment system and promoting remote diagnosis and treatment [60], so as to reduce excessive clustering in third-class hospitals. In addition, for high-risk buildings such as hospitals, defensive space design should be enhanced. On the other hand, the fragility of the food supply during the pandemic was also a major problem faced by people. Lai et al. proposed the idea of restoring self-sufficient urban agriculture, which was not applicable to Chinese cities [7]. Swiggy (similar to Uber Eats in the US), an Indian food delivery platform, used its advantages to ensure food supply for the vulnerable population in Bangalore [61]. Online group purchases and unified distribution via community management were two of China’s successful experiences in fighting the pandemic. In long-term planning and urban management, it is also necessary to consider the virtualized online operation of daily public service facilities such as markets, redefine the layout of the urban logistics system, and build a non-contact distribution and supply guarantee system for daily living materials. Public transportation also increased the risk of infection. In Wuhan, empirical evidence showed that places where bus and metro stations were densely distributed were accompanied by a high degree of disease transmission, among which the positive impact of bus stops was greater, with a coefficient of 0.321. Consequently, measures such as limiting the number of passengers per vehicle and increasing the number of bus or railway lines could be taken to reduce excessive crowds. At the same time, sufficient disinfection facilities should be arranged in public service facilities, as was shown in a nationwide survey in Spain, which could result in a greater willingness to use public transport in post-COVID-19 times [62]. Regarding modes of travel, residents need to be encouraged more to ride bicycles and walk. The pedestrian and bicycle lanes in Wuhan were relatively scattered and incomplete and hence a more connected and safer road network should be planned. For the university model, such as a city with nearly a hundred universities, the test in Wuhan suggested that the presence of universities was able to curb the spread of the disease to some extent [63], with a negative coefficient of −0.059, which demonstrated that it may be better to continue to maintain the closed management mode in China, since the experimental results proved that it had a positive impact on the fight against the pandemic.

Finally, high environmental quality can reduce the risk of infectious disease transmission. Studies have shown that newly built high-rise residential buildings in Wuhan had higher risks of infection than old multi-story residential buildings, with a negative coefficient of −0.100, and hence the study proposed that the proportion of future high-rise residential developments in cities should be appropriately reduced. On the other hand, the experimental results proved that better air quality, water distribution and green space accessibility had a positive impact on the fight against the pandemic in Wuhan, which shows that development managers should further focus on improving the quality of human settlements. In Australia, COVID-19 created more opportunities for people to come into contact with and learn about nature at home [13], while in China, the fact was that there were no adequate parks and open spaces where people in densely populated urban areas could walk [64,65,66], in spite of people’s growing interest in short walks [19,67]. Consequently, open public spaces, such as waters and green spaces, should be more balanced to guarantee fair accessibility for residents, since it was beneficial to the residents’ health [68,69], and was able to play a barrier function in the urban ecological space to enhance the city’s natural immunity and allow residents to live in a fair and resilient city environment.

### 5.3. Future Research and Limitations

This study provides a basic research paradigm for the influence mechanism of urban elements on COVID-19 at various stages of the pandemic development in urban spaces. Firstly, in general, in the early stage of the pandemic, as evidenced by this study, different cities around the world, with the same and different population volumes, could establish their own factor system that may affect disease transmission according to the common (density, public service facilities, open space, etc.) and unique attributes (mountain cities, lake cities, university cities, etc.) of the city, as well as from the three aspects of population density, human mobility and environmental quality, so as to identify related urban factors and help urban planners optimize the allocation of public service facilities, as well as rethinking the reasonable size and structure of the city. Secondly, with the mutation of the virus, the infectivity continued to increase and people entered the stage of normalization of social activities and pandemic prevention. The pandemic data became more accurate, transparent, standardized and refined, which helps researchers achieve refined exploration of the interaction between the spatial variation of disease and related factors in different urban regions; this also makes it possible to provide urban planners with more accurate and effective planning suggestions in the context of pandemic prevention.

Moreover, the research can be expanded on in the following aspects in the future: (1) Combined with more detailed flow data, a detailed agent model can be built for deep research of cities’ internal connections during the pandemic. (2) The model can be applied to the study of explorations of relationships between other chronic and acute diseases and cities.

The limitations of our data and analysis can be split into three aspects. Firstly, due to the distinct temporal character of the Weibo data, the effective sample size was only 729, and hence there was still missing disease information that involved privacy issues. Although the sample size was limited, the spatial coverage of the data was good, and the study applied Kernel Estimate Density (KDE) to generate surfaces of the pandemic in the main urban area of Wuhan, which could clearly show the difference in the level of disease prevalence between regions. However, the use of spatial estimates of Kernel Density Estimation (KDE) could cause potential bias in the regression analysis, which was non-evaluative and unavoidable in this study. Secondly, the study attempted to assess the bias of the Weibo help data by calculating the rate of participation in the helpline among those infected with COVID-19, while due to the limited availability of the public data, only the participation rate at the district level was obtained. This meant we were unable to prove that on a more granular scale, within each TAZ unit, people infected with COVID-19 had the same attitude to posts. Thirdly, the internal built-up environment of residents was not taken into account in the model. In general, this study can be regarded as proposing an idea based on limited and innovative data. If detailed data can be obtained, then more detailed correlations can be drawn in the same way.

## 6. Conclusions

During the outbreak of the pandemic, there was spatial variability of COVID-19 in urban spaces, and the identification of the variation, as well as their interaction mechanisms with urban elements, is of great significance in the fight against viruses and the future development of cities. The study contributes by proposing the linear regression model with innovative social media data and seventeen explanatory factors to achieve the quantitative study of the spread of COVID-19 within Wuhan city. Weibo case data provided urban scholars a unique opportunity to revisit the result of the natural development of COVID-19 in Wuhan city during the early stage of the pandemic, which was the main advantage compared to other city-level data. The data had a good sample coverage, which provided an opportunity to obtain disease information on a fine scale (such as TAZ unit level), since, even now, there is no way to obtain publicly accurate data in the early stage on a fine scale or district scale. Fourteen significant urban factors—the Euclidean Distance (ED) to the third ring road, markets, third-class hospitals, rivers and green space, the Kernel Density Estimation (KDE) values of the elderly population, middle schools, universities, bus stops, metro stations and waters, the Inverse Distance Weighted (IDW) value of air quality index (AQI), house prices, and the age of buildings—were finally selected to help us understand the distribution pattern of the COVID-19 in the urban space. Three characteristics of the areas of case aggregation were identified and the study further discussed the impacts and proposed detailed and targeted recommendations on the intensive management of high-density population and susceptible population areas, configuration and transformation of public facilities, and comprehensive improvement of urban environmental quality, to contribute to the construction of the healthy city which could well protect people from disease threats when faced with similar emergencies in the future.

## Figures and Tables

**Figure 1 ijerph-19-05208-f001:**
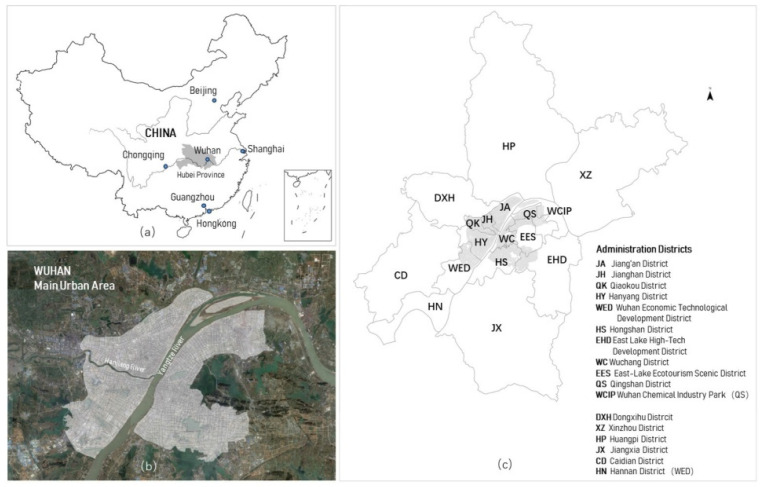
Map of the study area in Wuhan, China: (**a**) the geographic location of the Wuhan, China; (**b**) the main urban area of Wuhan (MUA); and (**c**) administrative districts of Wuhan [8].

**Figure 2 ijerph-19-05208-f002:**
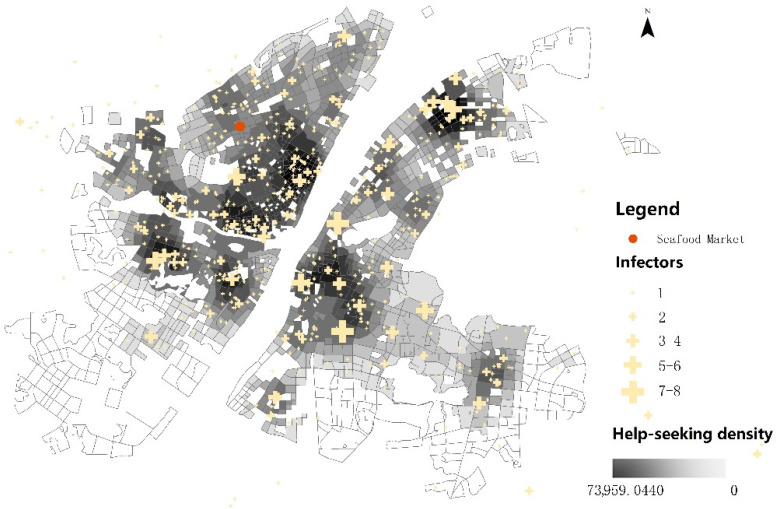
Density map of Weibo help data.

**Figure 3 ijerph-19-05208-f003:**
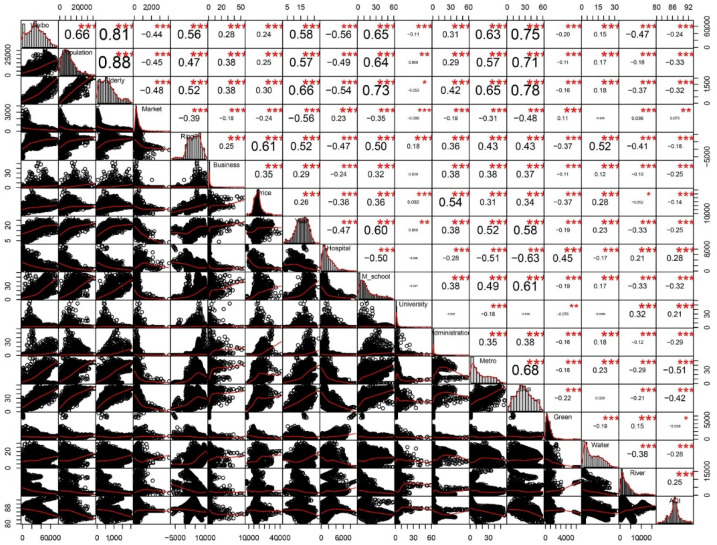
Variable scatter plot. Note: ***, **, * represent significant at the level of 0.001, 0.01, 0.05, respective.

**Figure 4 ijerph-19-05208-f004:**
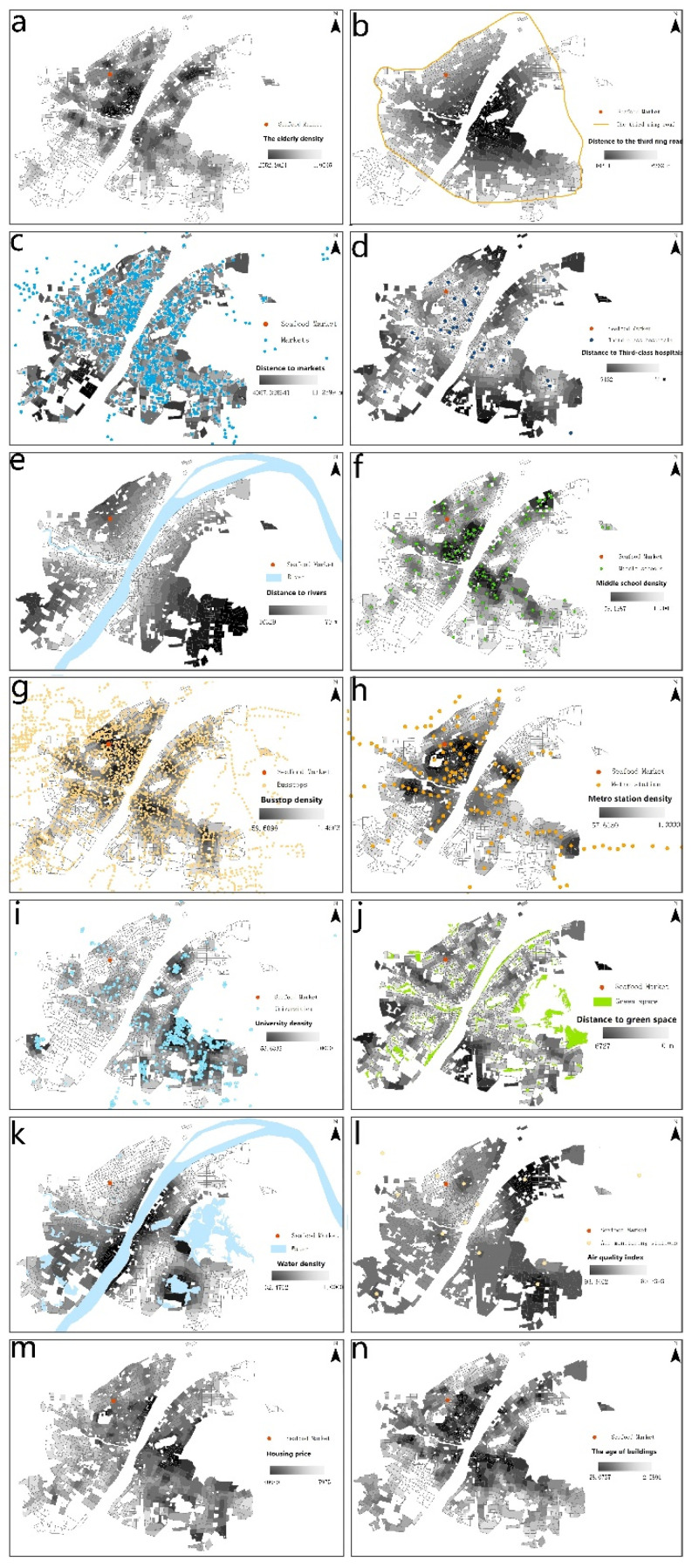
Spatial distributions of explanatory variables. (**a**) The elderly density; (**b**) Distance to the third ring road; (**c**) Distance to rivers; (**d**) Middle school density; (**e**) Distance to markets; (**f**) Distance to Third-class hospitals; (**g**) Bus stop density; (**h**) Metro station density; (**i**)University density; (**j**) Distance to green space; (**k**) Water density; (**l**) Air quality index; (**m**) Housing price; (**n**) The age of buildings.

**Figure 5 ijerph-19-05208-f005:**
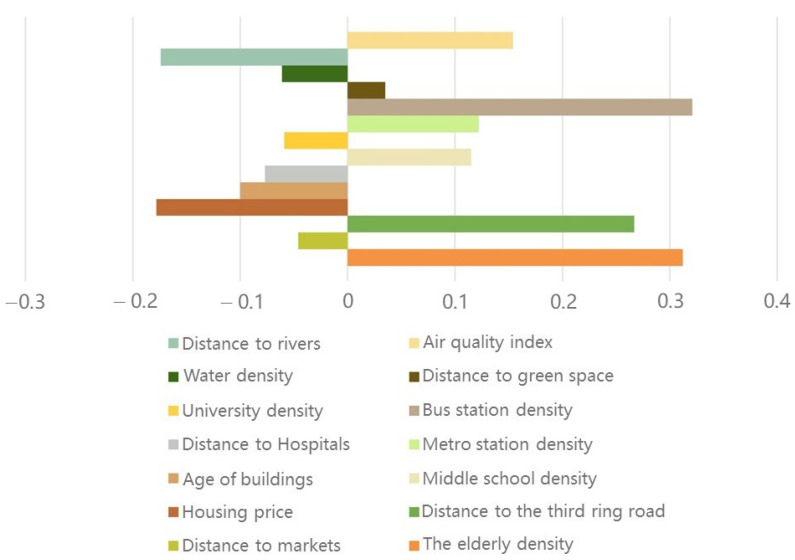
Statistics chart of standardized coefficients.

**Table 1 ijerph-19-05208-t001:** Case statistics of Wuhan.

Districts	Cumulative Confirmed Case	Proportion
Jiang’an (MUA)	6563	12.987%
Jianghan (MUA)	5242	10.373%
Qiaokou (MUA)	6854	13.562%
Hanyang (MUA)	4691	9.282%
Wuchang (MUA)	7551	14.942%
Qingshan (MUA)	2804	5.548%
Hongshan (MUA)	4720	9.340%
Dongxihu	2637	5.218%
Caidian	1424	2.818%
Jiangxia	875	1.731%
Huangpi	2117	4.189%
Xinzhou	1071	2.119%
East Lake Ecotourism Scenic District	483	0.956%
East Lake High-Tech Development District	2174	4.302%
Wuhan Economic Technological Development District	1108	2.192%
Other places	223	0.441%
Total	50,537	100.00%

As of 16 March 2022 source: Wuhan Municipal Commission of Health.

**Table 2 ijerph-19-05208-t002:** Statistics of onset period.

Date of Illness	Number of Weibo Help Information
20–30 Novenmber 2019	3
1–31 Januaru 2020	558
1–6 February 2020	154
7–10 February 2020	14

**Table 3 ijerph-19-05208-t003:** The participation rates in the helpline among the COVID infected.

Districts	Infections in Weibo Information	Cumulative Confirmed Case	the Participation Rate
Jianghan (MUA)	75	5137	1.46%
Qiaokou (MUA)	128	6789	1.89%
Wuchang (MUA)	135	7431	1.82%
Jiang’an (MUA)	145	6521	2.22%
Hanyang (MUA)	116	4661	2.49%
Hongshan (MUA)	87	4652	1.87%
Qingshan (MUA)	64	2773	2.31%

**Table 4 ijerph-19-05208-t004:** Variable selection.

Dimension	Variables	Definitions
Dependent variable	Weibo help case density (Weibo)	The average KDE ^1^ value of COVID-19 infectors in each unit
Population density	Population density (Population)	The average KDE ^1^ value of the population in each unit
	The elderly population density (Elderly)	The average KDE ^1^ value of the elderly population in each unit
	Distance to the third ring road (Ring3)	The average ED ^2^ value from units to the third ring road
	Distance to rivers (River)	The average ED ^2^ value from units to rivers
Human mobility	Distance to markets (Market)	The average ED ^2^ value from units to markets
	Distance to third-class hospitals (Hospital)	The average ED ^2^ value from units to hospitals
	Middle school density (M_school)	The average KDE ^1^ value of middle schools in each unit
	University density (University)	The average KDE ^1^ value of universities in each unit
	Business density (Business)	The average KDE ^1^ value of business facilities in each unit
	Administration density (Administration)	The average KDE ^1^ value of administration facilities in each unit
	Bus stop density (Bus)	The average KDE ^1^ value of bus stop in each unit
	Metro station density (Metro)	The average KDE ^1^ value of metro station in each unit
Environmental quality	Housing price (Price)	The average house price in each unit
	Age of buildings (Year)	The average age of buildings in each unit
	Air quality index (AQI)	The average IDW ^3^ value of air quality index in each unit
	Distance to green spaces (Green)	The average ED ^2^ value from units to green spaces
	water density (water)	The average KDE ^1^ value values of waters in each unit

^1^ KDE: The Kernel Density Estimation (see Methods for detailed information); ^2^ ED: Euclidean Distance; ^3^ IDW: The Inverse Distance Weighted (see Methods for detailed information).

**Table 5 ijerph-19-05208-t005:** Variable statistics.

Variable Name	Number	Minimum	Maximum	Mean	Std. Deviation
Weibo help density (Weibo)	1681	0.0000	73,959.0440	28,405.5334	17,269.6339
Population density (Population)	1681	2.5756	31,293.7673	8199.1254	5694.6717
The elderly population density (Elderly)	1681	1.9645	2052.2024	653.7139	464.9645
Distance to the third ring road (Ring3)	1681	–6662.0000	10,621.0000	4499.3998	3162.0162
Distance to rivers (River)	1681	70.0000	16,329.0000	3259.2570	3170.4919
Distance to markets (Market)	1681	11.2599	4087.3986	410.5552	421.3228
Distance to third-class hospitals (Hospital)	1681	23.0000	9432.0000	1913.4747	1535.0498
Middle school density (M_school)	1681	1.0000	59.1887	15.1634	13.7686
University density (University)	1681	1.0000	58.6333	3.6523	5.2843
Business density (Business)	1681	1.0000	54.7805	3.2398	5.1925
Administration density (Administration)	1681	1.0000	58.9231	4.2478	7.3873
Bus stop density (Bus)	1681	1.4563	59.6098	27.6569	13.4843
Metro station density (Metro)	1681	1.0000	57.6180	17.0827	14.5484
Housing price (Price)	1681	7975.0000	40,989.0000	18,447.5544	3879.5026
Age of buildings (Year)	1681	2.6891	28.0737	15.9859	4.5786
Air quality index (AQI)	1681	80.3393	93.3462	86.6823	2.1607
Distance to green spaces (Green)	1681	0.0000	6727.0000	662.9863	633.4966
water density (water)	1681	1.0000	32.4762	9.8986	7.4350

**Table 6 ijerph-19-05208-t006:** Model summary of seventeen urban factors.

Model	Unstandardized Coefficients		Standardized Coefficients	*t*-Value	Significance	Collinearity Statistics	
	Beta	Std. Error	Beta			Tolerance	VIF
(Constant)	−70,623.886	10,753.331		−6.568	0.000		
Population	−0.475	0.083	−0.157	−5.725	0.000	0.164	6.113
Elderly	16.697	1.28	0.450	13.041	0.000	0.103	9.714
Ring3	1.539	0.115	0.282	13.379	0.000	0.276	3.625
River	−0.781	0.088	−0.143	−8.914	0.000	0.473	2.114
Market	−2.144	0.607	−0.052	−3.531	0.000	0.558	1.794
Hospital	−0.923	0.186	−0.082	−4.978	0.000	0.45	2.223
M_school	151.174	21.843	0.121	6.921	0.000	0.403	2.479
University	−156.239	44.016	−0.048	−3.55	0.000	0.674	1.483
Business	−18.321	43.511	−0.006	−0.421	0.674	0.715	1.399
Administration	25.647	35.631	0.011	0.72	0.472	0.527	1.899
Bus	417.361	29.146	0.326	14.32	0.000	0.236	4.234
Metro	142.755	21.209	0.120	6.731	0.000	0.383	2.610
Price	−0.886	0.079	−0.199	−11.268	0.000	0.392	2.552
Year	−426.759	67.602	−0.113	−6.313	0.000	0.381	2.626
AQI	1132.903	119.915	0.142	9.448	0.000	0.543	1.840
Green	1.032	0.369	0.038	2.796	0.005	0.667	1.499
Water	−120.487	34.619	−0.052	−3.48	0.001	0.551	1.816
R: 0.893	R Square: 0.797		Adjusted R Square: 0.794				
Std. Error of the Estimate: 7831.238353							

**Table 7 ijerph-19-05208-t007:** Model summary of fourteen urban factors.

Model	Unstandardized Coefficients		Standardized Coefficients	*t*-Value	Significance	Collinearity Statistics	
	Beta	Std. Error	Beta			Tolerance	VIF
(Constant)	−80,667.568	10,629.470		−7.589	0.000		
Elderly	11.599	0.867	0.312	13.371	0.000	0.229	4.364
Ring3	1.457	0.114	0.267	12.750	0.000	0.286	3.501
River	−0.945	0.083	−0.174	−11.403	0.000	0.540	1.853
Market	−1.868	0.608	−0.046	−3.070	0.002	0.568	1.762
Hospital	−0.870	0.186	−0.077	−4.673	0.000	0.456	2.194
M_school	144.758	22.042	0.115	6.567	0.000	0.405	2.471
University	−193.204	43.907	−0.059	−4.400	0.000	0.692	1.444
Bus	411.545	29.433	0.321	13.983	0.000	0.237	4.226
Metro	144.609	21.162	0.122	6.833	0.000	0.393	2.543
Price	−0.793	0.069	−0.178	−11.471	0.000	0.519	1.928
Year	−378.367	66.530	−0.100	−5.687	0.000	0.402	2.490
AQI	1229.175	117.094	0.154	10.497	0.000	0.582	1.717
Green	0.942	0.373	0.035	2.527	0.012	0.668	1.496
Water	−140.884	34.832	−0.061	−4.045	0.000	0.556	1.799
R: 0.890	R Square: 0.792		Adjusted R Square: 0.790				
Std. Error of the Estimate: 7915.456643							

## Data Availability

Data sharing not applicable.

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
