# Peer review of "Correlation Analysis between Urban Elements and COVID-19 Transmission Using Social Media Data"

_ijerph, 2022, doi:10.3390/ijerph19095208_

Round 1

Reviewer 1 Report

Dear authors,

Thank you for reviewing the manuscript and accepting the corrections.

Author Response

Dear reviewer 1:

Really thanks for your helpful comments.

Reviewer 2 Report

The authors have replied to all the questions I raised in my review of their paper, but some of my most serious concerns are not cleared by their answers and comments. I try to summarize what is still not clear, at least from my point of view.

  • The nature of the data used in the paper, more precisely the use of social media data for estimating the impact of Covid-19 in each observation unit (TZA). What the authors write on this issue deal with my comment on the short time interval characterizing their data, but not with the more fundamental question related to the fact these data originate by a voluntary choice of individuals, posting their message to the helpline. There is no a priori reason to believe that people infected with Covid in the different TZA units had the same attitude to post messages. This was the reason why I asked whether the author had an idea of the participation rates to the helpline in the different TZA, while the authors make a computation of these rates for the Districts, which are not however their unit of analysis. I still maintain my concern about the potential bias that the nature of the data may introduce in the estimation of the Covid contagion in the different TZA, then used as dependent variable in the subsequent regression exercise.
  • A related issue is the one dealt with in the authors’ reply, at point 6. Their explanation does not address my concern, which is about the potential bias of their regression analysis introduced by the fact that they are not using values from the actual distribution of the variable of interest, but estimated values (with a nonparametric methodology). This point still needs clarification, looking at the econometric issue.
  • Even if the (most serious) technical issues are clarified, it remains the question about the nature of this study, which is not a spatial study, but simply an attempt of an econometric analysis of the determinants of the Covid transmission, using a different “space” from the ones used in other published work. The authors actually admit this in their reply to point 5, even if they still continue to use terms like spatial variations or spatial effects, while they correctly do not mention now spatial heterogeneity.
  • As for the reply to point 2, while the authors offer some attempt to justify the use of Euclidean distances, it is still not clear why for others they use the Kernel densities.
  • As for the reply to point 3, I do not understand how the use of population densities of TZA should respond to my question about the mass of people attracted by some facilities.
  • As for the reply to point 9, my comment was about the weak relation between the discussion on policy recommendations (section 5.2) and the results of the authors’ empirical analysis.

Author Response

Dear reviewer 2:

Really thanks for your helpful and detailed comments. We believe that we have been able to address your comments and suggestions, and that our paper has been substantially improved with your help. Please see the attachment.

Reviewer 3 Report

First of all, I wish to congratulate the authors for conducting such a pertinent and important study. A more nuanced understanding of city-level spatial factors that shape a region’s COVID-19 transmissions could help society better understand more precise ways to control pandemics. Overall, this study is of high impact and could help enrich the literature and inform better policymaking. Please find my comments below and address them properly, as I believe that, while they are minor in nature, shedding light on these concerns could help the authors further enhance the quality of their work and the readers better appreciate the study.

In the abstract section, the authors mentioned, “…this study explored the correlation of different infectious outcomes by COVID-19 transmission and 15 various factors of urban environment in the main urban area of Wuhan”, which is different from the study’s focus indicated in the title—"spatial variations of COVID-19 infections”. Please revise this, or elaborate on why “spatial factors” and “urban environment” are interchangeable concepts in this study. This is important as the significance of urban environment factors has been extensively discussed later in the manuscript.

Out of all the Weibo posts on the subject matter, which could be in the millions, why only 1,200 messages were selected? What is the screening process adopted to reduce the included posts to 729? How representative are these posts with regard to residents living in Wuhan?

Please provide a clear definition of “participation rates”. On a related note, how were “participation rates” calculated? Arguably, most Weibo users are young people/people with a smartphone. How was this factor adjusted to more realistically reflect the makeup of the population?

The authors mentioned that, “we selected seventeen explanatory variables from perspectives of…”. Are these factors selected based on theoretical frameworks? If not, what is the rationale for considering these factors in the study? This is one of my key concerns, as this issue matters to the replicability of the study for future research endeavors. 

More precise language should be utilized across the manuscript. For instance, the authors mentioned, “Wuhan, the city where the first outbreak of the epidemic occurred.” Please consider using words like “identified” or “reported”, as opposed to “occurred”. Based on current understanding, it is unclear with regard to where did the first outbreaks occur in China or elsewhere in the world. The same need for precision goes for similar definitive statements, such as the one on Huanan wet market.

Could the authors please add 1-2 sentences to contextual Wuhan for the general audience? Such as the size of the population and the city’s economic/transportation significance in Hubei Province or across China? This could help the readers better appreciate the findings and significance of the study.

It seems that, in the discussion/conclusion section, how the findings of this study could benefit future research (e.g., global cities with similar urban structures) is barely discussed. Please add a paragraph to further elaborate why this study matters not only in the current pandemic but future ones as well.

The manuscript could benefit from additional rounds of editing. For instance, the authors mentioned, “Secondly, the main challenge hampering the research of spatial correlation between epidemics and urban spatial factors was the lack of the fine spatial information of epidemics infectors…”, without offering specific examples of “fine spatial information”. Please address this issue, along with other similar ones, accordingly.

Author Response

Dear reviewer 3:

Really thanks for your helpful and detailed comments, which has made our article greatly improved. We have checked the manuscript and revised it according to the comments. We submit here the response to your comments. Please see the attachment.

Round 2

Reviewer 2 Report

The new version of the work surely includes revisions that effectively take into account some of the comments I made on previous versions. This is the case, for instance, of section 5.2 (see point 5 in the authors' cover letter). However, the most serious doubts I raised in my previous comments still survive. As for point 1 in the authors' cover letter, even if they now acknowledge the limitations of their data in the paper, I still consider that these limitations a serious flaw for the significance of their analysis and, above all, of their results. On point 2, I just take note of the different points of view with the authors about the spatial nature of their analysis, since I am still convinced that their empirical exercise does not add much to other works carried out at a different spatial level. As for point 3, there is an improvement, insofar the authors provide a justification for using Kernel estimates for some variables, but still they do not catch my point about the need to explain the different treatment of different variables (i.e. why Kernel or Euclidean cannot be used for all the variables). As for point 4, there is an appreciable explanation of the different weights for the different variables, but it is not what I was asking for. 

Author Response

Dear reviewer 2: 
Really thanks for your helpful and detailed comments, which has made our article 
greatly improved. We have checked the manuscript and revised it according to the 
comments. Please see the attachment.

This manuscript is a resubmission of an earlier submission. The following is a list of the peer review reports and author responses from that submission.

Round 1

Reviewer 1 Report

The paper provides an interesting investigation in the early stage of the pandemic by COVID-19 in the Wuhan city. The authors present substantial evidences that could help us to understand the dissemination of COVID-19 and how we can deal to avoid future outbreak. I highlighted a few adjusts to the paper.

General Comments
# Authors, please check the space between the parenthesis and the text. Review all the
body of the manuscript. You need to standardize.
Line 36: “Studies” must be tiny: “studies”.
Line 44: Please check the space before “Besides”.
Line 56: Please check the expression ([6,21)].
Line 90: Is it “phrases” the correct term?
Line 234: Check the space in the expression “figure2”. Please review all body when
figures and tables appear.
Line 255: Please start the phrase “fourteen” whith uppercase “Fourteen…”
Line 364-366: Please do not end the paragraph with “:”

Reviewer 2 Report

The paper’s main objective, as stated by the authors, is the exploration of the correlation between COVID-19 transmission and spatial heterogeneity represented by various factors of urban environment, in the main urban area of Wuhan, using multiple regression. The results show that several factors are statistically significant, and the authors try to discuss their implications for policy makers. The empirical strategy of the paper is characterized by the use of social media data, for the assessment of COVID-19 diffusion at an intra-urban level; kernel density functions, for the estimation of the density of the values of different variables in the intra-urban units of observation (Traffic analysis zones, TAZ); a linear multiple regression model, for the assessment of different factors characterizing the different urban areas of Wuhan.

While the research objective of the paper is sound, I have some reservations about the data and the methods employed by the authors, which, so far, cast serious doubts about the actual achievement of what they are aiming at.

As for the data, the use of social media data is not adequately motivated and discussed. They are used for estimating the incidence of COVID-19 in each observation unit (through the Kernel density function). However, one may wonder what sort of bias they introduce in the estimation, once considering: 1) the very short period of observation (according to the authors, “the Weibo help information was mainly distributed from February 3rd to February 12th, and most of the information was posted from February 4th to 8th”) and 2) the fact that people voluntarily posted their messages. In other words, not only we do not know whether these data are a reliable source of information for estimating the extent of contagion in Wuhan (given the short period of time and the subjective participation to the help line of the social media), but also, we cannot be sure of whether they reflect real differences in the pattern of contagion across the different units of observation (the TAZ zones), which is a very critical issue for the achievement of the main objective of the paper. Is there any way to be sure that, for instance, the participation rates to the help line of the social media among the Covid infected is homogenous in all the TZA units? At least, the issue needs to be discussed, and the authors have to convince readers of the lack of any substantial and significant bias in using these data. Some problems may also arise from the use of the independent variables, which should represent mobility (distance to markets and third-class hospitals, and estimated densities related to middle schools, universities, business, administration, bust stop and metro station). It is not clear, first, why for some of these variables the authors use (Euclidan) distances while for others they estimate (Kernel) densities. Second, while the existence and concentration of different human activities, like the ones relative to the different variables used in the paper, is surely a determinant of contagion, what is relevant, also in terms of differences across the TZA units, is the mass of people that these activities bring together: it will sure make a difference whether a school in one unit has ten students enrolled or one thousand. Finally, a minor point related to the identification of the observation unit of the study. The authors present the study area, with a specific subsection (2.1), showing the subdivision of the urban area in Wuhan in the administrative districts, and including data relative to the number of contagions in these districts. At a very first reading of the paper, I was induced to think that the districts were the observation units, also because the TZA were mentioned very shortly in the presentation of the data. Then, if the authors wish to maintain a (sub)section presenting the study area, I strongly recommend including the presentation of TZA in this (sub)section, with more details on the characteristics of this specific subdivision of the urban area and, possibly, a discussion contrasting the administrative districts and the TZA units.

As for the methodology, one of the main critical issues I envisage in this study, is the use of the term spatial heterogeneity, which is not consistent with the technical definitions provided in the spatial analysis “mainstream”. As it can be clearly read in the works of one of the leading figures of this literature, spatial heterogeneity identifies one of the spatial effects (the other one being spatial dependence), which is characterized by “structural instability, either in the form of non-constant error variances in a regression model (heteroskedasticity) or in the form of variable regression coefficients” (L. Anselin, Spatial Econometrics, in B.H. Baltagi (ed.), Spatial Econometrics, Blackwell Publishing, 2003, 310-330, p. 310). The definition has implications for the specification of spatial heterogeneity in the empirical analysis (on the point see L. Anselin, Thirty years of spatial econometrics, Papers in Regional Science, 89(1), 2010, 3-25, as well as other works from the same author). I believe that the authors are simply testing the marginal effect of different variables on the Covid-19 density, exploiting the variability of these variables across the TAZ zones of Wuhan, but they do not actually deal with the potential variability of their marginal effects across the TZA zones, as driven by an observable and, above all, unobservable heterogeneity of the different spatial units. As a consequence, the results of the paper can be hardly conceived of as an assessment of how “the mechanism of the urban environment” affects “the spatial distribution of COVID-19”, which appears to be its main objective of research. At most, these results can be regarded as an assessment, at the urban level, of the impact of different factors, already examined at a different geographical level by previous studies. Another serious issue is related to the use of the Kernel functions. First of all, there is no motivation of why the authors use the kernel estimates (based on their observed values) in the regression, and why they prefer these estimates to the observed values themselves or to other potential transformation of these values (e.g., the logarithmic values). A convincing discussion of this issue is vital, for the authors to sell their results. Second, since the densities can be estimated with very different (Kernel) functions and different bandwidth rules, the authors should discuss the selection of their function and of their bandwidth. Third, I have some doubts about the values in Table 3, which probably need some clarification. The average value of the Kernel estimate should be a conditioned average of the values of the variable of interest. How is it possible, then, that when we consider the dependent variable (the Weibo help density), with a total of 729 messages (for all the city) we get an average value of 28,405.5334? And what about the range of values for this variable? I wonder how any transformation of the observed values can give these estimates. Leaving aside the values of the estimates, another doubt, requiring clarification, is about the use of the Kernel estimates as regressors. A common problem in regression analysis is that when the observed values are substituted by other values related to their distribution like, for instance, the mean, to avoid bias in the estimation, correction factors need to be introduced. Are we sure that the simple use of Kernel estimates as regressors does not introduce similar biases? The authors should include convincing references that this problem is under control, in their analysis.

Finally, the results in Tables 4 and 5 are poorly commented on, and the discussion carried out in section 5 is then barely related to these results, appearing more as a general assessment of the relevance of the independent variables in the COVID-19 diffusion.

Reviewer 3 Report

First of all, I wish to congratulate the authors for conducting a relevant and insightful study that could help society better understand how to control and prevent current as well as future pandemics. An in-depth understanding of the interplay between spatial factors and COVID-19 transmission could shed light on how infections cluster in urban areas. Overall, this study is of high impact and could help enrich the literature and inform better policymaking. Please find my comments below and address them properly, as I believe that addressing these concerns is necessary to help the authors further strengthen their paper and the readers better appreciate the study. Line 57-58, how is “intra-urban scale” defined? In the same vein, what is the definition used for “administrative region level” or “urban elements”? Please provide detailed definitions of these concepts, as the authors build their “so what” question upon these terms. The authors chose 17 urban factors for their analysis. How were these factors selected? If theoretical frameworks were used, which ones? Also, how are “urban factors” defined? How does it differ from or similar to the concept of “geospatial factors?”. This is important because, without an established link (e.g., theoretical connection) that could shed light on why these 17 factors are studied, as opposed to other equally important ones (e.g., car registration in the area), the key argument of this study could be hardly founded. A related concern I have is that the authors have adopted so many terms (e.g., “environmental quality”) without providing a detailed explanation in terms of what do they entail, why are they influential to the study, or why should the readers learn about them. This could substantially hinder the readers’ ability to fully appreciate the study. Please consider revising the manuscript thoroughly to address this issue. Only 1,200 Weibo messages were selected, and Wuhan has over 11 million people. What makes the author believe that 1,200 messages are enough to represent a city as big as Wuhan? If well-evidenced rationales were used that could back up this decision, please share them in detail in the manuscript. Also, what is the selection criteria used for using these messages, as opposed to others? Overall, detailed elaborations are needed to ensure that the method section is sound enough to answer the research question.